

# The *p53* signaling pathway of the large yellow croaker (*Larimichthys crocea*) responds to acute cold stress: evidence via spatiotemporal expression analysis of *p53, p21, MDM2, IGF-1, Gadd45, Fas,* and *Akt*

Baoying Qian, Xin Qi, Yi Bai and Yubo Wu

School of Life Science, Taizhou University, Taizhou, Zhejiang, China

Corresponding author
Baoying Qian, wu-tongye1979@126.com

## ABSTRACT

The *p53* activation is induced by stressors, such as DNA damage, oxidative stress, and activated oncogenes, and can promote cell cycle arrest, cellular senescence, and apoptosis. The large yellow croaker (*Larimichthys crocea*) is an important warm temperate marine fish in the Chinese aquiculture industry. However, few studies have investigated the role of *p53* in the response of *L. crocea* to environmental stressors. Therefore, the aim of the present study was to assess the spatiotemporal mRNA expression levels of genes involved in the *p53* signaling pathway of the large yellow croaker in response to cold stress. The results showed significant changes in the expression levels of *p53*, *p21*, *MDM2*, *IGF-1*, *Gadd45*, *Fas*, and *Akt* in various tissues of the large yellow croaker in response to cold stress for different times. As compared to the control group, *p53* mRNA expression was upregulated in most of the examined tissues at 24 h with the exception of the gill. In the liver, the expression levels of *p53* and *Fas* were significantly decreased at 12 h, while those of *p21*, *MDM2*, *IGF-1*, *Gadd45* were dramatically increased. *Akt* expression was notably changed in response to cold in several tissues. These results suggested that *p53* was potentially a key gene in the large yellow croaker response to cold and possibly other environmental stressors.

## INTRODUCTION

Net cage culturing of the large yellow croaker (*Larimichthys crocea*) is economically important to the marine aquaculture industry in China (*Liu & Han, 2011*). Over the past decade, low temperature resistance of *L. crocea* has gained considerable attention because of extensive economic losses caused by cold stress in winter, especially in the East China Sea (*Xu, Chen & Ding, 2012*). In order to breed large yellow croaker with stronger resistance to low temperatures and a lower death rate during the winter season, previous studies have investigated serum expression levels of physiological and biochemical markers,
antioxidant production, enzymatic activities, and the proteome in response to cold stress, which showed that the serum biochemical indicators glutamate pyruvate transaminase, glutamic oxaloacetic transaminase, and alkaline phosphatase were significantly affected (*Ji et al., 2009*; *Li et al., 2010*; *Zhang et al., 2013*). More recent studies have focused on genes in response to cold stress. Liver transcriptome analysis indicated that the expression levels of numerous genes were either up- or down-regulated after 12 h of cold stress (*Qian & Xue, 2016*). Moreover, the expression profile of cold-inducible RNA-binding protein was significantly changed in different tissues of the large yellow croaker during acute cold stress (*Miao et al., 2017*). Also, *Li, Ran & Chen (2018)* found that *MIPS* mRNA expression was significantly up-regulated in gill, heart, muscle and brain, and indicated that *MIPS* may participate in response to acute or chronic cold stress. As the molecular mechanisms underlying the activation of these genes are complicated, further studies are needed to fully understand the genetic responses of the large yellow croaker to cold stress.

The tumor suppressor *p53* not only plays key roles in the inhibition of cell carcinogenesis and tumor development, but also promotes cell cycle arrest and apoptosis (*Levine & Oren, 2009*; *Kastenhuber & Lowe, 2017*), and is involved in autophagy modulation, homeostatic regulation of metabolism, pluripotency, and repression of cellular plasticity (*Aylon & Oren, 2016*). On account of the negative regulation of *MDM2* (induced by *p53*), basal levels of *p53* are low. The *p53* gene acts as a "guardian of the genome" under physiological conditions (*Momand et al., 1992*; *Haupt et al., 1997*; *Honda, Tanaka & Yasuda, 1997*; *Kubbutat, Jones & Vousden, 1997*) and is activated by stress signals, such as DNA damage, oncogene activation, and environmental stress. However, the response of *p53* is dependent on the intensity of the stress signal, the cell type, and the stage of cellular differentiation (*Horn & Vousden, 2007*; *Kastenhuber & Lowe, 2017*). Notably, the *p53* response is exceedingly flexible, as even a very low basal level of *p53* can protect the cell from the accumulation of DNA damage and subsequent carcinogenesis, which under different stress signals occurs through two typical mechanisms: (1) the promotion of cell senescence and apoptosis in response to severe or constant genotoxic and cellular stressors, and (2) the promotion of temporary cell cycle arrest in order to maintain cell survival prior to DNA repair (*Jones et al., 2005*; *Moddocks et al., 2013*; *Chen et al., 2016*; *Pappas et al., 2017*).

Various target genes of the *p53* signaling pathway involved in the arrest of cellular growth have been investigated, which include growth arrest and DNA damage-inducible protein (*Gadd45*), cyclin-dependent kinase inhibitor 1A (*p21*), and tumor necrosis factor receptor superfamily member 6 (*Fas*) (reviewed in *Levine & Oren, 2009*). The results of our previous study showed that the *p53* signaling pathway was significantly enriched in the liver of the large yellow croaker in response to cold stress for 12 h, while numerous genes related to cell cycle arrest, apoptosis, and DNA repair and damage prevention were remarkably affected (*Qian & Xue, 2016*). Reportedly, *p53* promotes apoptosis in the gills of the Nile tilapia (*Oreochromis niloticus*) and zebrafish (*Danio rerio*) in response to cold stress (*Wang, 2016*). Similarly, *p53* mRNA expression was significantly upregulated in the muscle tissue of *D. rerio* under low temperature stress (*Li et al., 2018*).

We found that the *p53* signaling pathway was enriched significantly in our previous study (*Qian & Xue, 2016*). Is the *p53* signaling pathway potential pathway in response to

acute cold stress? Basis on our previous result, the spatiotemporal expression of various genes (i.e., *p53*, *p21*, *MDM2*, *IGF-1*, *Gadd45*, *Fas*, and *Akt*) in different tissues of the large yellow croaker under cold stress were investigated in this study. The objectives of this study were (1) to investigate the role of partial genes involved in the *p53* signaling pathway in response to cold-induced stress in the large yellow croaker, and (2) to discover the changes of mRNA expression in different tissues of the large yellow croaker under different acute cold stress time.

## MATERIALS AND METHODS

### Ethics statement
The study protocol was approved by the Medical Ethics Committee of Taizhou University Medical College (TZYXY2019-211) and conducted in accordance with the guidelines of the Institutional Animal Care and Use Committee. To minimize suffering, all experiment fish in this study were anesthetized by Tricaine-S (TMS, MS-222)(50 mg /L) firstly. And then we collected the tissues of large yellow croaker after the fish lost consciousness. Lastly, the fish which has been taken a surgery would be sprayed with anesthetic (200 mg/L), and let it death with euthanasia.

### Fish and induction of acute cold stress
A total of 180 large yellow croakers (mean weight, $80 \pm 0.7$ g) were purchased from a mariculture farm located in Xiangshan Bay (Zhejiang Province, China), randomly assigned to one of six groups (30 fish/group), and then cultured in 500-L plastic aerated tanks in the laboratory of the Ningbo Marine and Fishery Science and Technology Innovation Base (Zhejiang Province, China) for 7 days. During the adaption period, the fish were fed granulated feed twice per day (04:30 and 18:30 h). A total of 90 fish cultured in three tanks were exposed to acute cold stress with the use of ice wrapped in thick plastic bags until the water temperature decreased to 14 °C within 2 h (cold stress group), while the other 90 fish in three tanks were cultured at environmental temperature and received no treatment (control group). After 1, 3, 6, 12, 24, 48, and 72 h of acute cold stress, the liver, muscle, gill, heart, spleen, intestine, brain, and kidney were collected from three fish in the cold stress group and control group, respectively, and immediately snap-frozen in liquid nitrogen, then stored at −80 °C.

### Total RNA extraction and cDNA synthesis
Total RNA was extracted from each sample using the E.Z.N.A.® Total RNA Kit I (Omega Bio-Tek, Inc., Norcross, GA, USA) in accordance with the manufacturer's instructions. Total RNA was quantified with a NanoDrop™ 1000 Spectrophotometer (NanoDrop Technologies, LLC, Wilmington, DE, USA) and RNA integrity was assessed with the use of an Agilent 2100 Bioanalyzer (Agilent Technologies, Inc., Santa Clara, CA, USA). The RNA integrity value of all samples was ≥ 8. The extracted RNA was stored at −80 °C until analysis.

First stand cDNA was synthesized from total RNA using the PrimeScript™ RT reagent Kit with gDNA Eraser (Takara Bio, Inc., Kusatsu, Shiga Prefecture, Japan) in accordance

**Table 1  Primers for quantitative real time PCR.**

| Gene | Primer sequence (5′—3′) | Gene | Primer sequence (5′—3′) |
|---|---|---|---|
| β-actin | F: TCGGTATGGAATCTTGCG | Fas | F: CACTCCAGCAGGGAAATGGA |
|  | R: GTATTTACGCTCAGGTGGG |  | R: GCCATTTTGCTACGTCTCGC |
| P53 | F: ACTACTGCCGGCCTAATGTG | Akt | F: TGCCCCAGCATGAATGAAGT |
|  | R: GCAAACTGCATGGTTGGAGG |  | R: GTTGTGGTCACTGGACACCT |
| MDM2 | F: TAGACGCCGTGCATGGATTT | Gadd45 | F: ATCAACGTGGTGCGAGTCAA |
|  | R: CCAGTTTGTTGTCATCGGCG |  | R: CATTGCAGTAGCGTGTGCAG |
| P21 | F: GGGAAATGGCACCAATGTCG | IGF-1 | F: GTTCATTTTCGCCGGGCTTT |
|  | R: GACGAAGAAGATGTCCGCCT |  | R: ACAGCACATCGCACTCTTGA |

with the manufacturer's instructions and stored at −20 °C until quantitative real-time polymerase chain reaction (qRT-PCR) analysis.

## Spatiotemporal expression analysis

The qRT-PCR analyses of the spatiotemporal expression profiles of *p53*, *p21*, *MDM2*, *IGF-1*, *Gadd45*, *Fas*, and *Akt* were conducted with primers designed using Primer Premier 5 software (Premier Biosoft, Palo Alto, CA, USA) (Table 1). β-actin was used as a housekeeping gene. Before qRT-PCR, the amplification efficiency of the primers was evaluated with five 10-fold serial dilutions of cDNA of all tissues.

The qRT-PCR analysis was performed using a CFX96 Real-Time PCR System (Bio-Rad Laboratories, Hercules, CA, USA) with a total reaction volume of 20 µL, consisting of 1 µL of cDNA diluted to 1:5 with sterile DNase/ RNase-free distilled water, 0.6 µL of the forward primer, 0.6 µL of the reverse primer, 9 µL of FastStart Universal SYBR Green Master mix (Sigma-Aldrich Corporation, St. Louis, MO, USA), and 8.8 µL of sterile DNase/RNase-free distilled water. The following thermal cycling conditions were used: 95 °C for 10 min followed by 40 cycles at 95 °C for 15 s, 58 °C for 20 s, and 72 °C for 20 s. A melting curve was generated. Each sample was amplified in triplicate and the relative expression levels of *p53*, *p21*, *MDM2*, *IGF-1*, *Gadd45*, *Fas*, and *Akt* were normalized to that of *β-actin* with the $2^{-\Delta\Delta CT}$ method (*Thomas & Livak, 2008*). Statistical significance was determined using one-way analysis of variance. All statistical analyses were performed using IBM SPSS Statistics for Windows, version 21.0. (IBM Corporation, Armonk, NY, USA). A probability (*p*) value of <0.05 was considered statistically significant and <0.01 as highly significant. All qPCR data could be obtained in "Supplemental File" which named with raw data.

## RESULTS

### Analysis of the *p53* signaling pathway

The results of our previous study showed that the expression profiles of genes involved in the *p53* signaling pathway were significantly affected and this pathway was remarkably enriched in the liver transcriptome of the large yellow croaker in response to acute cold stress for 12 h (*Qian & Xue, 2016*). Gene networks associated with cell cycle arrest, apoptosis, *p53* negative feedback, and DNA repair and damage prevention were evaluated in the present study based on the transcriptome data of previous studies (Fig. 1). In this
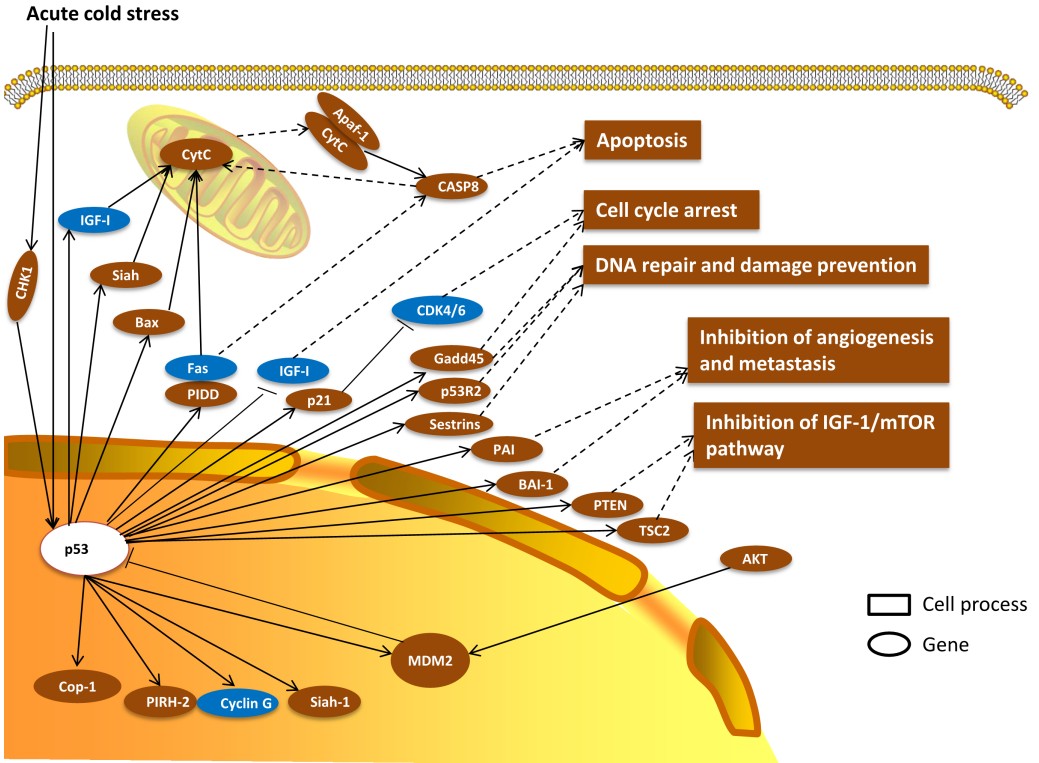

**Figure 1** **Putative gene networks in large yellow croaker stressed by 12 h acute cold based on the transcriptome data of previous studies.** Enriched gene networks associated with cell cycle arrest, apoptosis, p53 negative feedback, and DNA repair and damage prevention. Brown indicates up-regulated, blue indicates down-regulated, white indicates no changes. Full names of abbreviated genes are listed in Abbreviation.

gene networks, genes expression related to apoptosis were significant changed , including *Siah*, *Bax*, *CytC*, *PIDD*, *Apaf-1* and *CASP8* were increased significantly and *IGF-1*, *Fas* were decreased significantly. Genes related to cell cycle arrest such as *p21* and *Gadd45* expression were increased, while *CDK4/6* was decreased significantly. And the other genes expression of *p53R2*, *Sestrins* (related to DNA repair and damage prevention), *PAI*, *BAI-1* (related to inhibition of angiogenesis and metastasis), *PTEN*, *TSC2* (related to inhibition of IGF-1/mTOR pathway) were upregulated remarkably. In addition, downstream genes of p53 including *Cop-1*, *PIRH-2*, *Siah-1* were also significant increased.

## The mRNA profiles of Akt, MDM2, p53, p21, Gadd45, Fas, and IGF-1 in liver tissue

The mRNA expression profiles of genes and gene networks in the liver of the large yellow croaker in response to acute cold stress were investigated (Figs. 2A$_{1-7}$, 2B). Liver mRNA expression levels of *p53* were significantly decreased in response to cold stress at 3, 6 and 72 h, while significantly increased at 24 and 48 h. In addition, the mRNA levels of *Akt*, *MDM2*, *p21*, and *Gadd45* were significantly increased at 1 and 3 h, while the expression levels of *p21* and *Gadd45* were significantly upregulated with the exception of 48 h. There

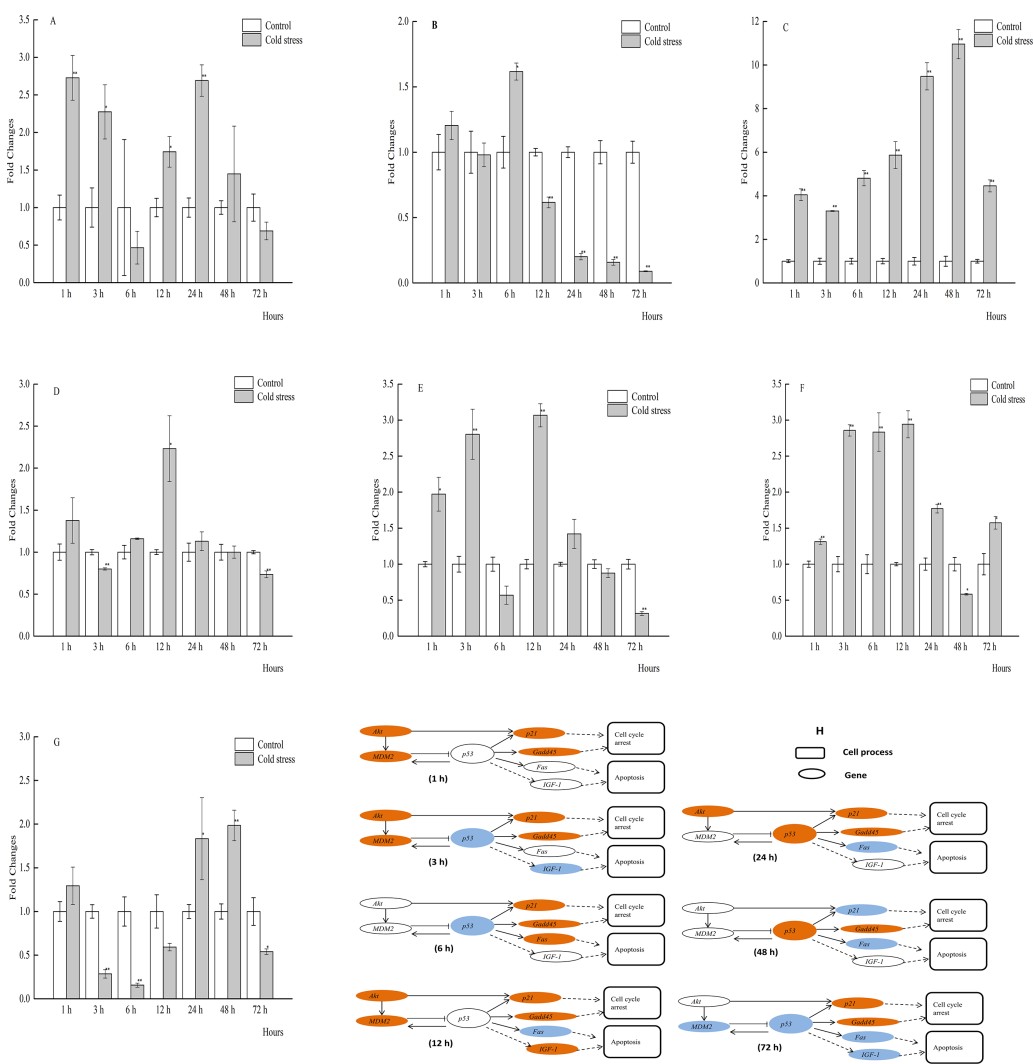

**Figure 2  qPCR analysis of genes in the liver of large yellow croaker under acute cold stress.** (A) *Akt* (protein kinase B), (B) *Fas* (tumor necrosis factor receptor superfamily member 6), (C) *Gadd45* (growth arrest and DNA damage-inducible protein), (D) *IGF-1* (insulin-like growth factor 1), (E)*MDM2* (E3 ubiquitin-protein ligase mdm2), (F) *p21* (cyclin-dependent kinase inhibitor 1A), (G) *p53* (tumor protein p53); (H) putative gene networks based on qPCR data; orange indicates up-regulated, blue indicates down-regulated, white indicates no changes. The results are expressed as mean fold change $\pm$ SD ($n = 3$ fish per treatment). Significant differences were considered at *$0.01 \leq P < 0.05$ and **$P < 0.01$.

was a good agreement between the qRT-PCR findings and previous liver transcriptome data of the large yellow croaker in response to acute cold stress at 12 h. The changes in mRNA expression levels of the selected genes were comparable between the two methods, although *IGF-1* expression was increased by qRT-PCR, while decreased according to the transcriptome data. The qRT-PCR results showed that the mRNA expression levels of *Akt*, *MDM2*, *p21*, and *Gadd45* were significantly increased, while that of *p53* was increased after

12 h of acute cold stress, as compared to the control group, although this difference was not statistically significant.

## Spatiotemporal expression patterns of Akt, MDM2, p53, p21, Gadd45, Fas, and IGF-1

The spatiotemporal expression profiles of *Akt*, *MDM2*, *p53*, *p21*, *Gadd45*, *Fas*, and *IGF-1* in sampled tissues of large yellow croaker were determined by qRT-PCR analysis. The results of qRT-PCR analysis are shown in Figs. 2A–2G and 2H (liver), 3A–3G and 3H (muscle), 4A–4G and 4H (brain), 5A–5G and 5H (spleen), 6A–6G and 6H (gill), 7A–7G and 7H (kidney), 8A–8G and 8H (intestine), and 9A–9G and 9H (heart). *p53* mRNA expression levels in muscle were obviously increased after 1, 6, 12, 24, and 72 h of acute cold stress, but were significantly decreased at 3 h. mRNA levels of *Gadd45* and *p21* were increased in muscle throughout the cold stress period, although *Gadd45* expression was downregulated at 3 h. The mRNA expression levels of *Akt* and *Fas* were downregulated, while those of *MDM2*, *p53*, *p21*, *Gadd45*, and *IGF-1* were upregulated in muscle tissue at 12 h (Figs. 3A–3H).

Brain mRNA expression levels of all selected genes were significantly affected after 1 and 48 h of acute cold treatment. Notably, *p53* expression was significantly increased at 1, 12, 24, and 48 h, but not at 3, 6, and 72 h. Also, the expression levels of *MDM2* and *Gadd45* were dramatically upregulated in the brain tissue of the large yellow croaker throughout most of the treatment period (Figs. 4A–4H).

In contrast to the brain tissue, *p53* mRNA expression in the spleen tissue was obviously increased at 24 and 48 h, but not at all times. Spleen mRNA expression levels of *Gadd45* were increased after cold stress, although there was no statistical significance at 72 h. In addition, the expression levels of *Akt*, *MDM2*, *p53*, *p21*, *Gadd45*, and *IGF-1* were significantly upregulated at 24 h, while that of *Fas* was significantly downregulated (Figs. 5A–5H).

In the gill tissues, *p53* expression was significantly downregulated at 1, 3, and 72 h, and significantly upregulated at 6 and 12 h. *MDM2* and *Akt* mRNA expression levels were obviously increased at 1, 24, 48, and 72 h, and at 3, 6, and 12 h of cold stress, respectively. *MDM2* mRNA expression was not significantly affected, while *Akt* expression was significantly decreased. Gill mRNA expression levels of *p21* were significantly increased throughout the cold stress period, as was that of *Gadd45* with the exception at 72 h (Figs. 6A–6H).

Kidney mRNA expression levels of *p21* and *Gadd45* were significantly increased throughout the acute cold stress period, while that of *Fas* was obviously decreased with the exception of 1 and 24 h. The mRNA expression levels of *p53* in the kidney tissues were significantly upregulated at 1, 6, 12, 24, and 48 h. *MDM2* mRNA expression was upregulated throughout the cold stress period, although there was no significant change at 6 and 12 h (Figs. 7A–7H).

The mRNA expression of *p53* in the intestinal tissue was significantly upregulated at 1, 24, and 48 h, but downregulated at 3, 6, 12, and 72 h. At 24 and 48 h after acute cold stress,

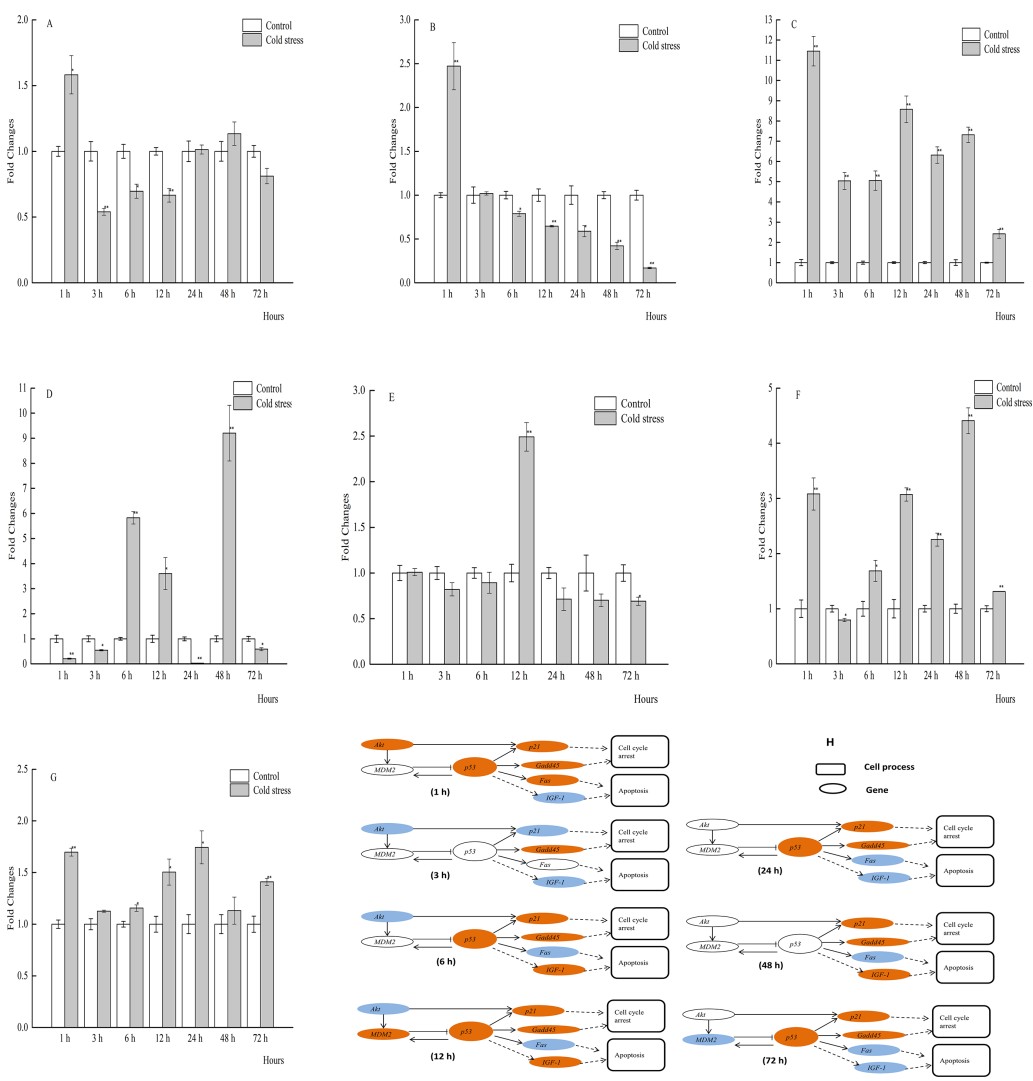

**Figure 3  qPCR analysis of genes in the muscle of large yellow croaker under acute cold stress.** (A) *Akt* (protein kinase B), (B) *Fas* (tumor necrosis factor receptor superfamily member 6), (C) Gadd45 (growth arrest and DNA damage-inducible protein), (D) *IGF-1* (insulin-like growth factor 1), (E) *MDM2* (E3 ubiquitin-protein ligase mdm2), (F) *p21* (cyclin-dependent kinase inhibitor 1A), (G) *p53* (tumor protein p53); (H) putative gene networks based on qPCR data; orange indicates up-regulated, blue indicates down-regulated, white indicates no changes. The results are expressed as mean fold change ±SD ($n = 3$ fish per treatment). Significant differences were considered at *$0.01 \leq P < 0.05$ and **$P < 0.01$.

the mRNA expression levels of *Akt*, *MDM2*, *p53*, *p21*, *Gadd45*, *Fas*, and *IGF-1* were all significantly increased. At 72 h, the mRNA expression levels of *Akt*, *MDM2*, *p21*, *Gadd45*, and *IGF-1* were significantly upregulated, while those of *p53* and *Fas* were downregulated (Figs. 8A–8H).

The qRT-PCR results indicated that in the heart tissues, the mRNA expression levels of *Akt*, *MDM2*, *p53*, and *Gadd45* were significantly increased, while *p53* was significantly down-regulated at 3 h with no significant difference at 1 and 12 h (Figs. 9A–9H).

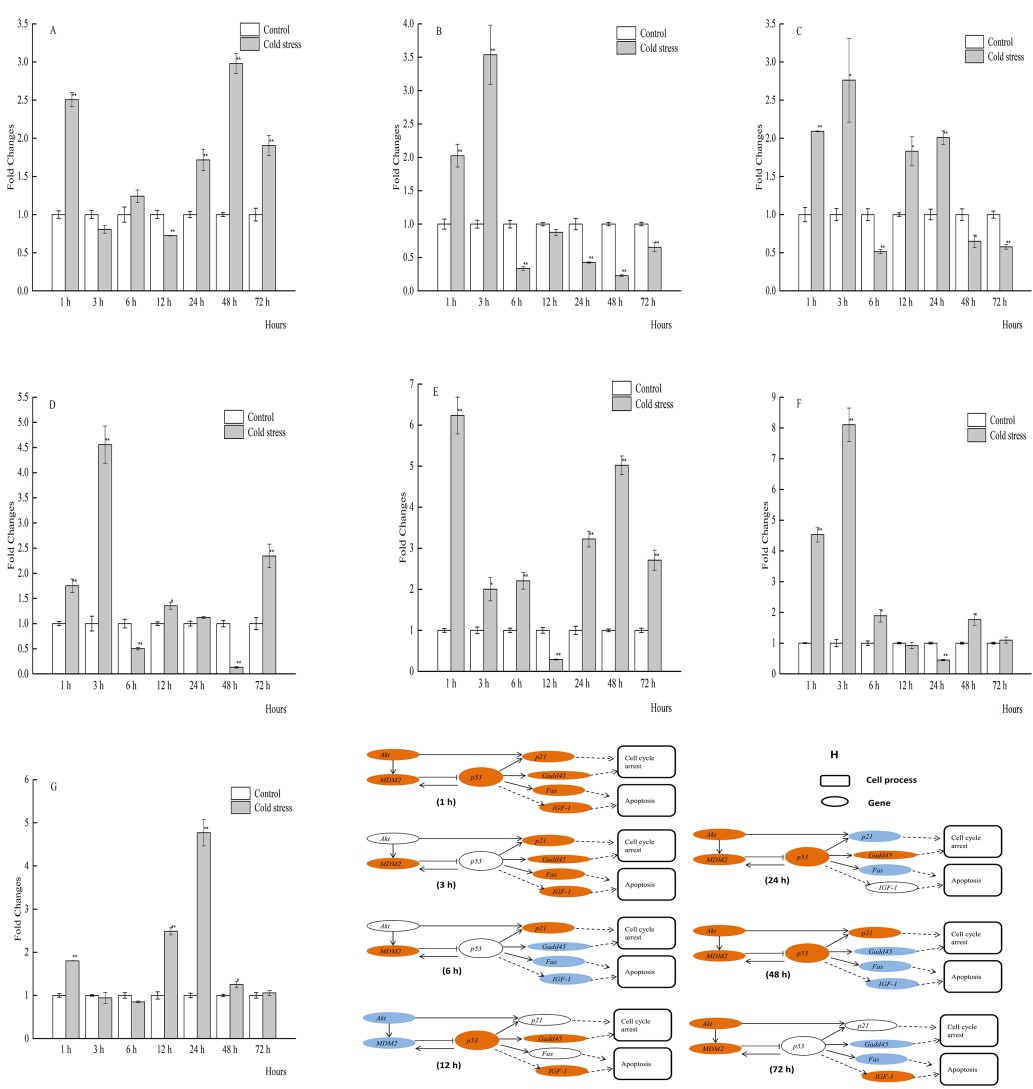

**Figure 4** **qPCR analysis of genes in the brain of large yellow croaker under acute cold stress.** (A) *Akt* (protein kinase B), (B) *Fas* (tumor necrosis factor receptor superfamily member 6), (C) *Gadd45* (growth arrest and DNA damage-inducible protein), (D) *IGF-1* (insulin-like growth factor 1), (E) *MDM2* (E3 ubiquitin-protein ligase mdm2), (F) *p21* (cyclin-dependent kinase inhibitor 1A), (G) *p53* (tumor protein p53); (H) putative gene networks based on qPCR data; orange indicates up-regulated, blue indicates down-regulated, white indicates no changes. The results are expressed as mean fold change $\pm$ SD ($n = 3$ fish per treatment). Significant differences were considered at *$0.01 \leq P < 0.05$ and **$P < 0.01$.

# DISCUSSION

Low temperatures that exceed the tolerance range of fish are known to disrupt energy metabolism, biochemical composition, immune function, and gene expression (*Lu et al., 2019*; *Song et al., 2019*). Net caged fish usually occur cold stress in winter, and the fish maybe suffer chronic cold stress in the sea while could suffer acute cold stress in the small waters. *Chen (2015)* found that there were similar changes between the mRNA expression of genes *MIPS*, *CIRP*, *SCD-1a* and *SCD-1b* in the tissues of large yellow croaker underlying

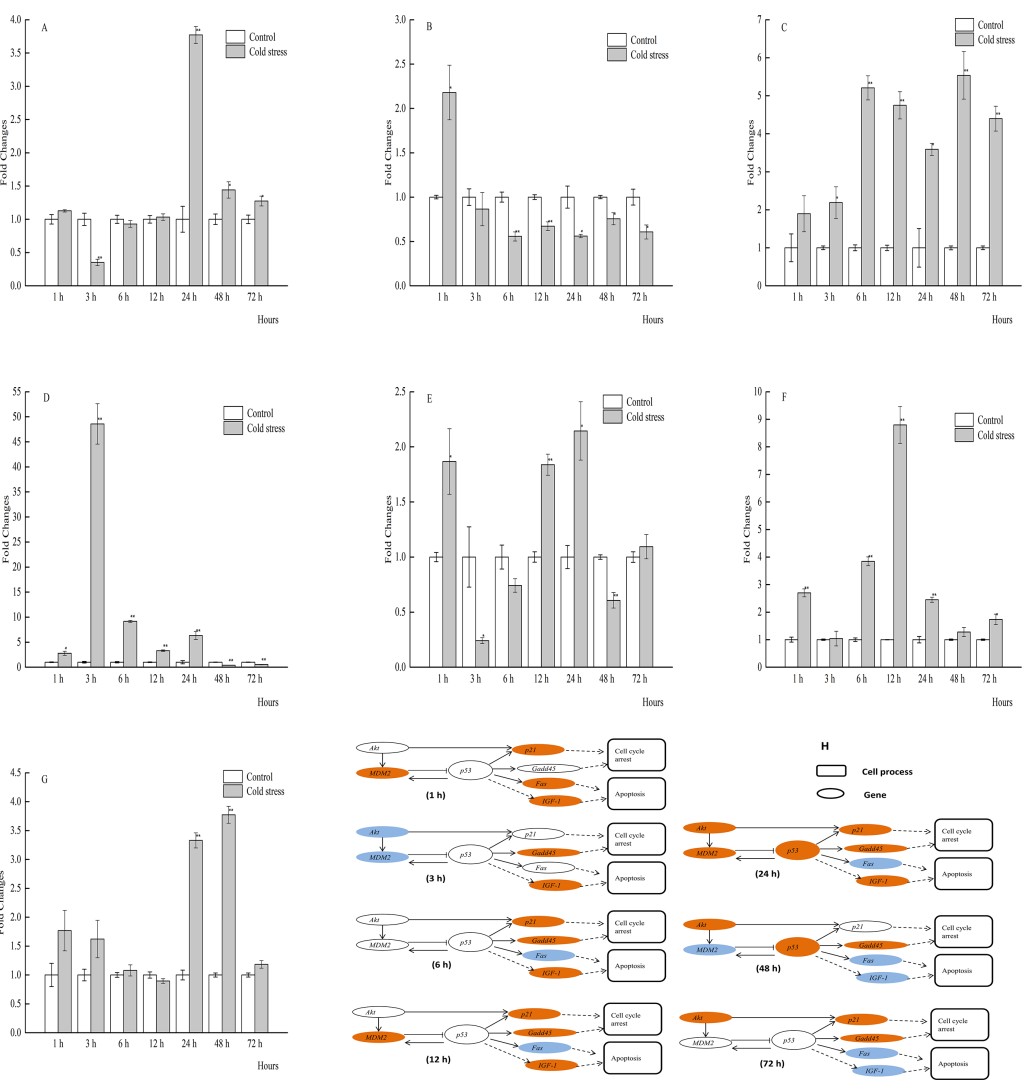

**Figure 5** **qPCR analysis of genes in the spleen of large yellow croaker under acute cold stress.** (A) *Akt* (protein kinase B), (B) *Fas* (tumor necrosis factor receptor superfamily member 6), (C) *Gadd45* (growth arrest and DNA damage-inducible protein), (D) *IGF-1* (insulin-like growth factor 1), (E) *MDM2* (E3 ubiquitin-protein ligase mdm2), (F) *p21* (cyclin-dependent kinase inhibitor 1A), (G) *p53* (tumor protein p53); (H) putative gene networks based on qPCR data; orange indicates up-regulated, blue indicates down-regulated, white indicates no changes. The results are expressed as mean fold change ± SD ($n = 3$ fish per treatment). Significant differences were considered at *$0.01 \leq P < 0.05$ and **$P < 0.01$.

chronic cold stress and acute cold stress. In this study, the mRNA expression of genes including *p53*, *p21*, *Fas* et al. were investigated in the 8 tissues of large yellow croaker occurred acute cold stress, the results could provide basic information for molecular mechanism in low temperature resistance in this fish. The results of our previous study of the liver transcriptome of *L. crocea* in response to 12 h of acute cold stress, identified a large number of differentially expressed genes that were enriched in the *p53* signaling pathway. Specifically, the mRNA expression levels of *MDM2*, *p21*, *Gadd45*, *CytC,* and *Apaf-1* were

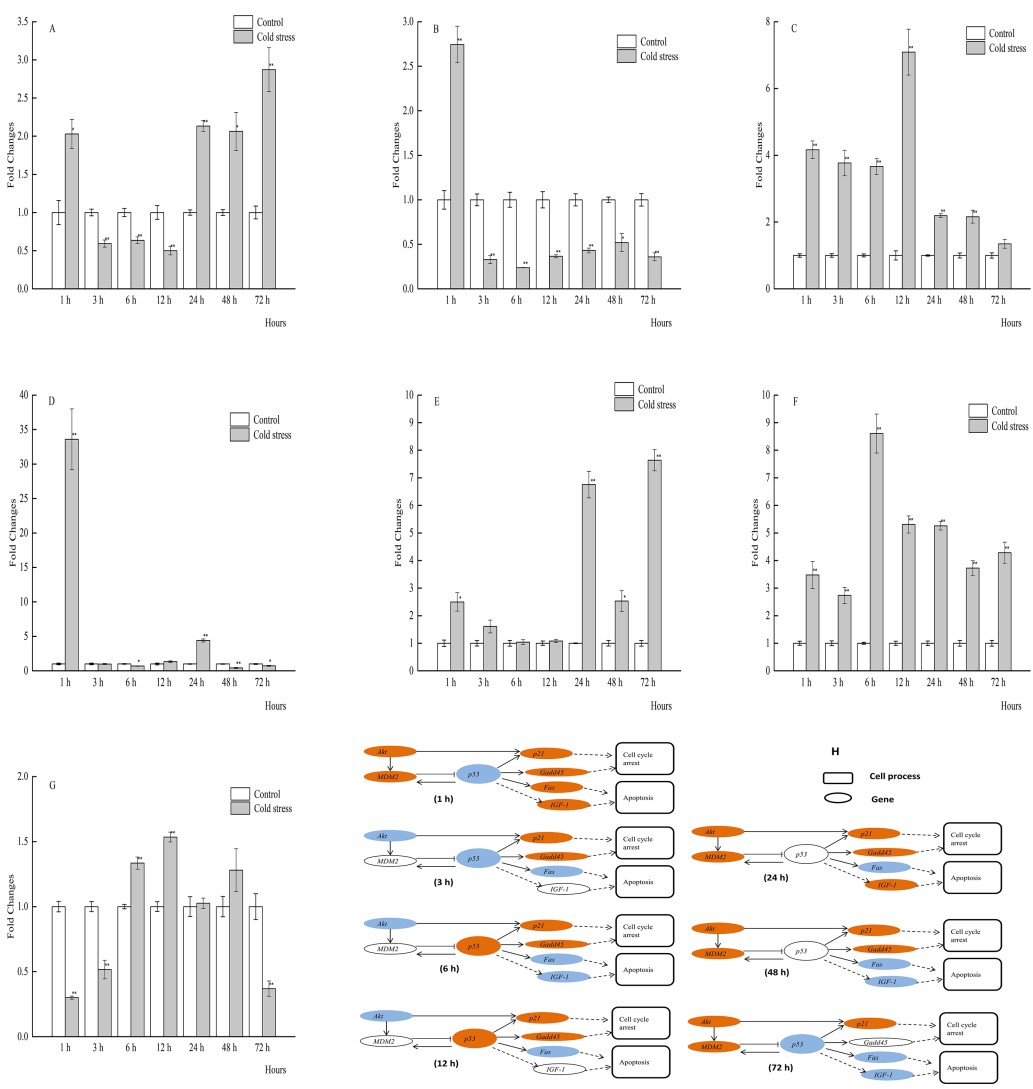

**Figure 6** **qPCR analysis of genes in the gill of large yellow croaker under acute cold stress.** (A) *Akt* (protein kinase B), (B) *Fas* (tumor necrosis factor receptor superfamily member 6), (C) *Gadd45* (growth arrest and DNA damage-inducible protein), (D) *IGF-1* (insulin-like growth factor 1), (E) *MDM2* (E3 ubiquitin-protein ligase mdm2), (F) *p21* (cyclin-dependent kinase inhibitor 1A), (G) *p53* (tumor protein p53); (H) putative gene networks based on qPCR data; orange indicates up-regulated, blue indicates down-regulated, white indicates no changes. The results are expressed as mean fold change $\pm$ SD ($n = 3$ fish per treatment). Significant differences were considered at $^*0.01 \leq P < 0.05$ and $^{**}P < 0.01$.

significantly increased, while those of *Fas*, *IGF*, and *CDK4/6* were significantly decreased. In addition, there was no significant change in the mRNA expression levels *p53* or other genes related to cell cycle arrest, apoptosis, inhibition of angiogenesis and metastasis, DNA repair and damage prevention, and *p53* negative feedback (*Qian & Xue, 2016*). In the present study, acute cold stress altered the expression profiles of genes related to cell cycle arrest and apoptosis in the liver, muscle, brain, spleen, gill, kidney, intestine, and heart. Of interest, the expression levels of *p21* and *Gadd45*, which are related to cell cycle arrest, were

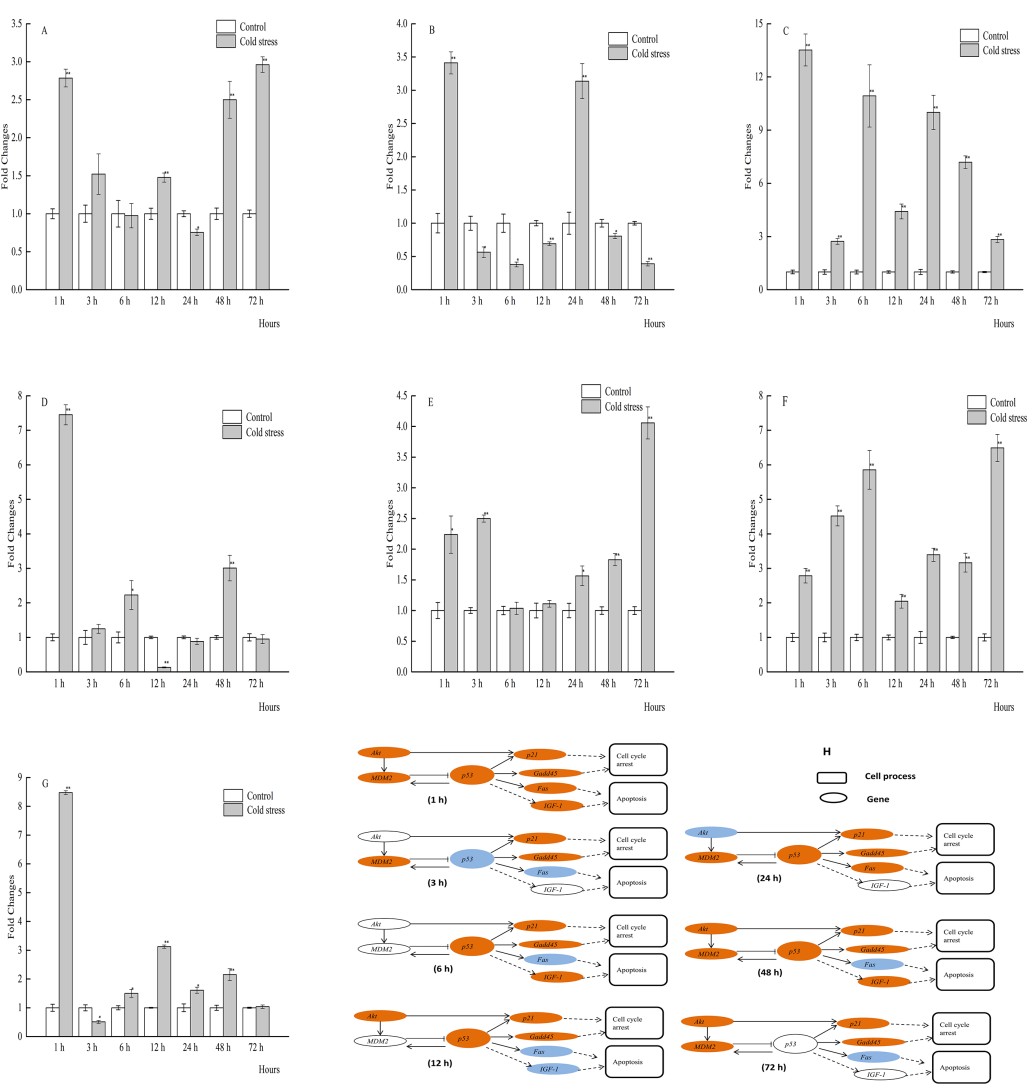

**Figure 7 qPCR analysis of genes in the kidney of large yellow croaker under acute cold stress.** (A) *Akt* (protein kinase B), (B) *Fas* (tumor necrosis factor receptor superfamily member 6), (C) *Gadd45* (growth arrest and DNA damage-inducible protein), (D) *IGF-1* (insulin-like growth factor 1), (E) *MDM2* (E3 ubiquitin- protein ligase mdm2), (F) *p21* (cyclin- dependent kinase inhibitor 1A), (G) *p53* (tumor protein p53); (H) putative gene networks based on qPCR data; orange indicates up-regulated, blue indicates down-regulated, white indicates no changes. The results are expressed as mean fold change $\pm$ SD ($n = 3$ fish per treatment). Significant differences were considered at *$0.01 \leq P < 0.05$ and **$P < 0.01$.

significantly changed in the liver, muscle, and kidney tissues throughout the cold stress period, while those of *Fas* and *IGF-1*, which are related to apoptosis, were also significantly altered in the heart tissue. One possibility was that there has tissue-dependence of the large yellow croaker response to acute cold stress. The tissues of liver, muscle and kidney were more sensitive to acute cold stress, and cell cycle arrest was influenced firstly in these tissues when the large yellow croaker occur cold stress. And to better adapt to the cold environment, apoptosis was firstly influenced in heart response to acute cold stress.

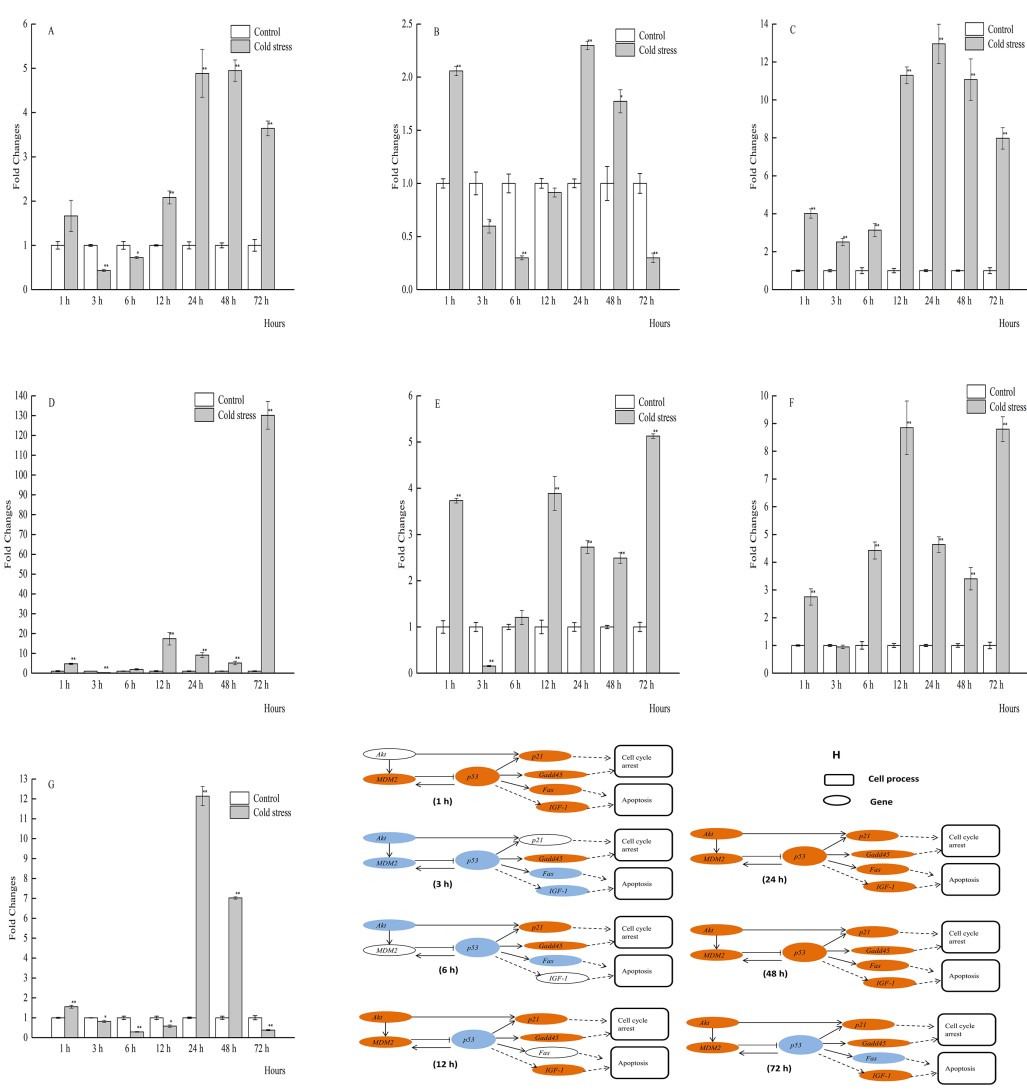

**Figure 8** qPCR analysis of genes in the intestine of large yellow croaker under acute cold stress. (A) *Akt* (protein kinase B), (B) *Fas* (tumor necrosis factor receptor superfamily member 6), (C) *Gadd45* (growth arrest and DNA damage-inducible protein), (D) *IGF-1* (insulin-like growth factor 1), (E) *MDM2* (E3 ubiquitin-protein ligase mdm2), (F) *p21* (cyclin-dependent kinase inhibitor 1A), (G) *p53* (tumor protein p53); (H) putative gene networks based on qPCR data; orange indicates up-regulated, blue indicates down-regulated, white indicates no changes. The results are expressed as mean fold change ±SD ($n = 3$ fish per treatment). Significant differences were considered at \*$0.01 \leq P < 0.05$ and \*\*$P < 0.01$.

As a guardian of the genome, *p53* is remarkably sensitive to environmental factors and is readily activated by multiple stress signals, especially in aquatic organisms in response to temperature change. *Li et al. (2018)* demonstrated that upregulation of *p53* expression in response to low temperature stress can cause tail malformation of the zebrafish. The molecular mechanism of the *p53* pathway in response to cold stress also involves *MDM2* (*Wang, 2016*; *Sun et al., 2019*). In this study, the mRNA expression levels of *p53* and *MDM2* were significantly increased in the brain tissue of the large yellow croaker after 1 h of acute

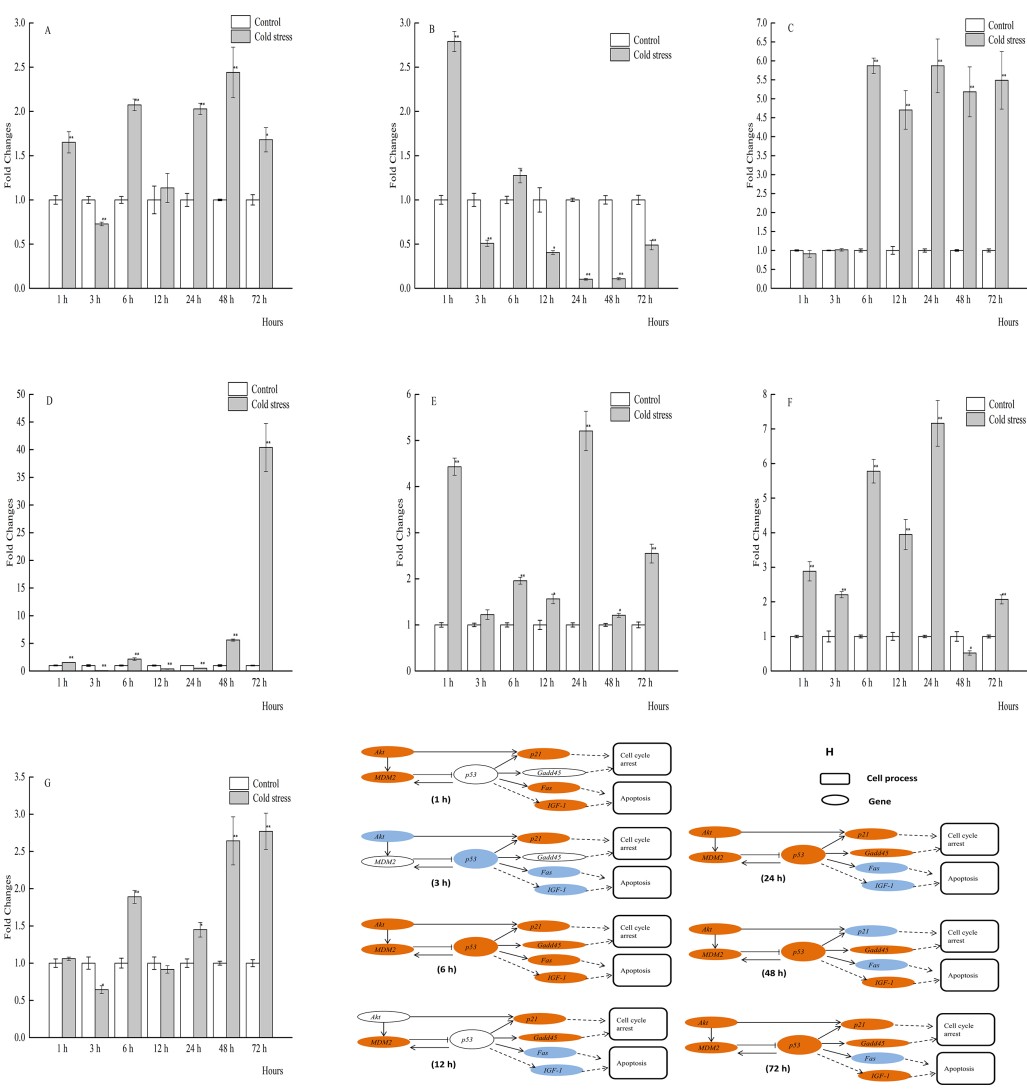

**Figure 9** **qPCR analysis of genes in the heart of large yellow croaker under acute cold stress.** (A) *Akt* (protein kinase B), (B) *Fas* (tumor necrosis factor receptor superfamily member 6), (C) *Gadd45* (growth arrest and DNA damage-inducible protein), (D) *IGF-1* (insulin-like growth factor 1), (E) *MDM2* (E3 ubiquitin-protein ligase mdm2), (F) *p21* (cyclin-dependent kinase inhibitor 1A), (G) *p53* (tumor protein p53); (H) putative gene networks based on qPCR data; orange indicates up-regulated, blue indicates down-regulated, white indicates no changes. The results are expressed as mean fold change ±SD ($n = 3$ fish per treatment). Significant differences were considered at *$0.01 \leq P < 0.05$ and **$P < 0.01$.

cold stress. Although there was no change in the *p53* expression profile, *MDM2* expression was increased at 3 and 6 h. *p53* induces the expression of *MDM2* and *MDM2* inhibits the activity and promotes the degradation of *p53* in a negative feedback loop (*Barak et al., 1993*; *Wu et al., 1993*; *Picksley & Lane, 1993*). In the present study, the expression levels of *p53* and *MDM2* in response to the same cold stress period differed among tissues. In contrast to that observed in the brain, muscle, kidney, and intestine, there was no significant change in *p53* mRNA expression in the liver after 1 h of cold stress, while *MDM2* mRNA expression
was significantly upregulated in the liver, heart, and spleen. There results may be due to the increased expression levels of *p53* in the liver (1.29-fold), heart (1.06-fold), and spleen (1.77-fold), which may have also impacted the expression of *MDM2* after 1 h of cold stress. In addition, the upregulation of *MDM2* inhibited *p53* expression in the liver at 3 and 6 h.

*Akt* plays key roles in glucose metabolism, apoptosis, cell proliferation, transcription, and cell migration. Activated *Akt* inhibits apoptosis through *MDM2* phosphorylation, which then inactivates *p53* (*Song, Ouyang & Bao, 2005*; *Farrel et al., 2009*). It has been reported that decreased mRNA expression of *Akt* and increased expression of *p53* can cause apoptosis of hepatocytes, suggesting an inverse correlation between these two genes (*Wu et al., 2016*). A delicate relationship between *Akt* and *p53* also occurred in the liver of the large yellow croaker after 1, 3, and 12 h of cold stress. During the cold stress period, the expression levels of *Akt* and *MDM2* were significantly increased, whereas *p53* expression was unchanged or decreased (Fig. 2B). The reason would be the up-regulated *Akt* induced the increased *MDM2*'s expression, and then inactivates *p53* in the liver of large yellow croaker under 1, 3, and 12 h of acute cold stress. In the present study *MDM2* expression was significantly increased at 12 h, and at other stress time, there have no significant changes in *MDM2 e* xpression (Fig. 3B). It is possible that the increased *p53* induced *MDM2* expression at 12 h, and at 3, 6 h stress time, *MDM2*'s mRNA expression were affected by the decreased *Akt* expression. In addition, it is not clear why *MDM2* expression has no significant changes while *Akt* and *p53* expression were both increased at 1 h stress and decreased significantly at 72 h. One possible could be there are other biological pathway or regulated genes affect the expression of these three genes. Furthermore, the relationships among *Akt*, *MDM2*, and *p53* seemed to be more intricate in different tissues in response to acute cold stress. *Akt* mRNA expression in the brain was significantly increased at 1 h, returned to normal levels from 3 to 6 h, decreased at 12 h, and increased again from 24 to 72 h, while *MDM2* expression was upregulated during most of the cold stress period, but not at 12 h (Fig. 4B). These results indicated that *MDM2* was regulated by *Akt* as well as *p53*.

*Fas* and *p21* are target genes of the *p53* pathway. The activation of p21 usually predicts the beginning of cell cycle arrest, whereas *Fas* promotes apoptosis. Previous studies have reported that short pulses of *p53* activity usually lead to cell cycle arrest, as the p21 promoter is more sensitive to this signaling output, while sustained *p53* signaling usually leads to changes in *Fas* expression, resulting in apoptosis. However, the mRNA signal of *p53*, either short or sustained, had no impact on the maximal level of the translated *p53* protein (*Espinosa, Verdun & Emerson, 2003*; *Gomes & Espinosa, 2010*; *Morachis, Murawsky & Emerson, 2010*; *Kastenhuber & Lowe, 2017*). In this study, hepatic mRNA levels of *p53* increased by 1.29-fold in response to cold stress at 1 h. Although this change was not statistically significant, this slight increase in *p53* expression could be sufficient to result in a change in *p21* expression. In addition, mRNA expression of *p21* was remarkably increased from 1 to 24 h, even at 72 h, although *p53* expression was decreased, while *Fas* expression was decreased or remained comparatively unchanged. It is not clear why there were such differences in the expression patterns of *p21* and *Fas*, as it seems that targeting of *p53* by these genes had no impact on the expression profiles of *p53* at 3, 6, 12, 48, and 72 h in the

liver tissues. Similar changes in expression levels occurred in the other tissues at certain times. One possibility could be complex regulatory signals from other regulatory proteins or pathway which regulate the expression levels of *p21* and *Fas* in response to acute cold stress. Hence, further studies are warranted to fully understand the molecular mechanism of the correlation to other unknown regulatory genes with *p21* and *Fas*.

*Gadd45* is as an important carcinogenic stress response factor that is sensitive to physiological and environmental stressors, and usually induced by cell cycle stagnation, DNA damage, and apoptosis (*Liebermann & Hoffman, 2008*; *Salvador, Brown-Clay & Fornace, 2013*; *Peng et al., 2015*). It has been observed that *Gadd45* can interact with *p21*, which activates *p53* via *p38* to maintain *p53* signaling (*Smith et al., 1994*; *Vairapandi et al., 1996*; *Azam et al., 2001*; *Liebermann & Hoffman, 2008*). In the present study, *Gadd45* was sensitive to both long- and short-term cold stress, as indicated by the consistently high expression levels in the liver, intestine, kidney, and muscle at all-time points. Even in the other tissues (spleen, heart, and gill), *Gadd45* mRNA expression was significantly increased at most time points. *Gadd45* is known to prevent DNA damage and promote DNA repair (*Peng et al., 2015*). It was possible that DNA damage was induced by acute cold stress, which resulted in significant increases in *Gadd45* expression. In addition, increased *Gadd45* expression may impact *p21* expression as mentioned previously.

*IGF-1* plays an important role in the growth and proliferation of cellular (*Handayaningsih et al., 2012*). The level of *IGF-1* mRNA expression of Nile tilapia (*Oreochromis niloticus*) were significantly increased when increased water temperature (*VeraCruz et al., 2006*). In our study, *IGF-1* mRNA expression were decreased significantly in muscle of the large yellow croaker at 1, 3, 24, and 72 h acute cold stress, and in the other except spleen, it has no regularities between *IGF-1* expression with stress time. We didn't know why there were such changes in *IGF-1* expression. One possibility could be there were other regulation protein influence this gene expression in these tissues of the large yellow croaker under acute cold stress. In spleen, the expression of *IGF-1* were increased significantly at 1 to 24 h, but significantly decreased at 48 and 72 h acute cold stress. The possible reason is that there were other regulation genes induce the *IGF-1* expression for compensatory growth of spleen cells at 1 to 24 h acute cold stress, and at 48 to 72 h, it maybe has beyond the tolerate cold stress time and resulted in the decrease of *IGF-1* expression.

## CONCLUSION

The results of the present study indicated that genes involved in the *p53* signaling pathway were largely affected by acute cold stress. There were significant changes in the mRNA expression levels of *Akt*, *MDM2*, *p53*, *p21*, *Gadd45*, *Fas*, and *IGF-1* in the liver, brain, muscle, gill, kidney, intestine, heart, and spleen in response to acute cold stress. *p53* target *p21* and *Gadd45*, which are involved with cell cycle arrest and were more sensitive to cold stress than *Fas*. mRNA expression of *Gadd45*, which is involved in DNA repair, was significantly increased in most of the studied tissues (liver, muscle, kidney, and intestine) in response to cold stress. The results of this study are in agreement with those of prior studies, which reported that genes involved in the *p53* signaling pathway could be affected by acute

cold stress. However, further studies are needed to elucidate the molecular mechanisms of genes in the *p53* signaling pathway that are activated by low temperature stress.

**Abbreviations**

| | |
|---|---|
| **p53** | tumor protein p53 |
| **Akt** | protein kinase B |
| **IGF-1** | insulin-like growth factor 1 |
| **CytC** | cytochrome c |
| **Apaf-1** | apoptotic protease-activating factor |
| **CASP8** | caspase 8 |
| **CHK1** | serine/threonine-protein kinase |
| **Siah** | E3 ubiquitin-protein ligase SIAH1 |
| **Bax** | apoptosis regulator BAX |
| **Fas** | tumor necrosis factor receptor superfamily member 6 |
| **PIDD** | leucine-rich repeats and death domain-containing protein |
| **p21** | cyclin-dependent kinase inhibitor 1A |
| **CDK4/6** | cyclin-dependent kinase 4/6 |
| **Gadd45** | growth arrest and DNA damage-inducible protein |
| **P53R2** | ribonucleoside-diphosphate reductase subunit M2 |
| **Sestrins** | sestrin 1/3 |
| **PAI** | plasminogen activator inhibitor 1 |
| **BAI-1** | adhesion G protein-coupled receptor B1 |
| **PTEN** | phosphatidylinositol-3,4,5-trisphosphate 3-phosphatase and dual-specificity protein phosphatase PTEN |
| **TSC2** | tuberous sclerosis 2 |
| **MDM2** | E3 ubiquitin-protein ligase mdm2 |
| **Siah-1** | E3 ubiquitin-protein ligase SIAH1 |
| **Cyclin G** | cyclin G1 |
| **PIRH-2** | RING finger and CHY zinc finger domain-containing protein 1 |
| **Cop-1** | E3 ubiquitin-protein ligase RFWD2 |

# ACKNOWLEDGEMENTS

We thank International Science Editing for language editing this manuscript.

## Funding

This project was funded by Zhejiang Provincial Natural Science Foundation of China (Grant No. LGN18C190007) and Taizhou science and technology project (Grant No. 1901ny09). The funders had no role in study design, data collection and analysis, decision to publish, or preparation of the manuscript.

## Grant Disclosures

The following grant information was disclosed by the authors:

Zhejiang Provincial Natural Science Foundation of China: LGN18C190007.

Taizhou science and technology project: 1901ny09.

## Competing Interests

The authors declare there are no competing interests.

## Author Contributions

- Baoying Qian conceived and designed the experiments, performed the experiments, analyzed the data, prepared figures and/or tables, authored or reviewed drafts of the paper, and approved the final draft.
- Xin Qi performed the experiments, prepared figures and/or tables, authored or reviewed drafts of the paper, and approved the final draft.
- Yi Bai performed the experiments, analyzed the data, authored or reviewed drafts of the paper, and approved the final draft.
- Yubo Wu performed the experiments, authored or reviewed drafts of the paper, and approved the final draft.

## Animal Ethics

The following information was supplied relating to ethical approvals (i.e., approving body and any reference numbers):

The Medical Ethics Committee of Taizhou University Medical College approved this research (TZYXY2019-211).

## Data Availability

The raw data, including the mRNA expression of p53, p21, Akt, Gadd45, IGF-1, MDM2 and Fas in the tissues of the large yellow croaker under cold stress, are available in the Supplemental Files.

## Supplemental Information

Supplemental information for this article can be found online at http://dx.doi.org/10.7717/peerj.10532#supplemental-information.

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
