# Peer review of "The p53 signaling pathway of the large yellow croaker (Larimichthys crocea) responds to acute cold stress: evidence via spatiotemporal expression analysis of p53, p21, MDM2, IGF-1, Gadd45, Fas, and Akt"

_PeerJ, doi:10.7717/peerj.10532_

## Round 0.1 · original submission · Major Revisions

Please address all the queries and recommendations of the reviewers and revise your manuscript accordingly.

Reviewer 1 ·

Basic reporting

There are two main issues to consider. The first being that there are 65 figures showing RT-cPCR data on different tissues at different time points following drop in temperature. The authors need to come up with a better way to present their data. It is not possible for the reader to go through these figures.
Secondly, all data are interpreted such that changes in levels of mRNAs are due to p53. But it has to be considered that also other mechanisms can control these genes. In some instances the correlation to p53 mRNA levels are poor. In this way, the different models proposed are quite complicated.

Experimental design

This seems to be ok. However, in this study they subject the animals to cold shock whereas in the sea the change in temperature is likely to be slow. This should be discussed.

Validity of the findings

The data look fine but the interpretations might be wrong.

Additional comments

Most important is to find a better way to present the data.

Reviewer 2 ·

Basic reporting

no comment

Experimental design

no comment

Validity of the findings

no comment

Additional comments

1. There is little grammatical mistakes. Data should be latest so it should be better to use recent references at lines 45 and 46 to provide latest justification for your study. There is also need to state scientific hypothesis. It should be influential to give a clear idea, why the research was carried out and the novelty of the manuscript.
2. Figures are relevant and well labelled but Author should cross check the graphs as graphical presentation of 3a1-7 and 3b (muscle), p53 and Gadd45 mRNA expression level at 3 h were not consistent. Similarly, recheck graphical presentation of MDM2 and Gadd45 (Fig. 4a1-7, b) in brain tissue as well as in gill tissues (6a1-7 and 6b) recheck p53, MDM2 and Akt mRNA expression levels.
3. Author should explain how the altered expression of genes play role in activation of p53 signaling pathways at lines 214 and 215.
4. Author should mention the reason behind the inhibited p53 expression in the liver at specific time points at lines 234 and 235.
5. From the perspective of relationship between p53 and MDM2, it is suggested that author should also explain the effect of IGF-1 in p53 signaling pathways? How it play role in acute cold stress?

Reviewer 3 ·

Basic reporting

no comments

Experimental design

no comments

Validity of the findings

no comments

Additional comments

Stress play key regulation of various metabolic process including DNA damaging and cell cycle progression. The paper my Qian et al describe the effect of cold induced stress on large yellow croakers which is economically important marine fish in specially china and around. The author previously reports the cold stress on p53 signaling and continued the study on gene network.

Qian et al., in present study report on upregulation and down regulation of various genes involved in cold stress, the studied by qRT-PCR analysis which gives additional evidence on cold induced stress studied before, The paper is clear and written nicely

Although expression levels were elevated and reported using qRT-QPCR, It would be great if additional study included, like- is the expression levels were also observed in protein level by doing western blotting on some on the key genes (p53 and MDM2!) regulation.

This experiments may be mostly suggestion- if authors are interested in understanding expression of many genes involved, they should perform microarray on samples, this gives clear regulatory pathway involved in cold stress.

Author did not provide explanation…why there is sudden decrease and increase in the level of expression (between 3- 6 hr) qPCR analysis of Akt in liver tissue.

MDM2 is ubiquitin ligase and regulates activity of p53, in a high expression of MDM2 why decrease in p53 activity was not see, is author role out possibility of other biological path way ?

why there is sudden increase in qPCR analysis of p21 in the muscle, whereas early and later time point show no significance difference (Fig 13)


legends reading show 9 fig by data presents 65 figures this need to be fixed
and this is confessing

the conclusive model needed for putative gene network in combined study

Thanks

---

## Round 0.2 · accepted · Accept

All critiques were addressed and the manuscript was revised accordingly. I am pleased to accept your work.

Reviewer 3 ·

Basic reporting

no comments

Experimental design

no comments

Validity of the findings

no comments

Additional comments

Qian et al., in the revised version responded well to prior critiques and manuscript improved lot for readability and clarity specially figures. I would expect future finding from this group as they promised in the rebuttal letter.